# Unpacking the Processes that Catalyzed the Adoption of Best Management Practices for Lowland Irrigated Rice in the Mekong Delta

Rica Joy Flor [1,*], Le Anh Tuan [2], Nguyen Van Hung [1], Nguyen Thi My Phung [3], Melanie Connor [1], Alexander M. Stuart [1], Bjoern Ole Sander [1], Helena Wehmeyer [1,4], Binh Thang Cao [5], Hardwick Tchale [5] and Grant R. Singleton [1,6]

[1] International Rice Research Institute, Los Baños 4031, Philippines; hung.nguyen@irri.org (N.V.H.); m.connor@irri.org (M.C.); alex@pan-uk.org (A.M.S.); b.sander@irri.org (B.O.S.); h.wehmeyer@irri.org (H.W.); grantrsingleton@gmail.com (G.R.S.)
[2] Independent Researcher, District 7, Ho Chi Minh City 72900, Vietnam; anh-tuan.le@my.jcu.edu.au
[3] Independent Researcher, My Phuoc Ward, Long Xuyen 880000, Vietnam; phungnguyen777@gmail.com
[4] Institute of Human Geography/Urban and Regional Studies, Department of Environmental Sciences, University of Basel, 4056 Basel, Switzerland
[5] World Bank, Hanoi 100000, Vietnam; tcao@worldbank.org (B.T.C.); htchale@worldbank.org (H.T.)
[6] Natural Resources Institute, University of Greenwich, Chatham Maritime, Kent ME4 4TB, UK
* Correspondence: r.flor@irri.org

**Abstract:** Vietnam is supportive of the transition to sustainable rice production in the Mekong Delta. The national program promoted best management practices for rice production through "1 Must Do and 5 Reductions" (1M5R). This review traces the technological development and uptake of 1M5R in national policies and by end-users. We highlight the outcomes from various policy-supported initiatives and unpack plausible pathways that generated the widespread adoption of 1M5R in eight provinces in the Mekong River Delta: at least 104,448 smallholder rice farmers were reached, and 1M5R practices adopted on 113,870 hectares. The scaling of 1M5R was enabled through a convergence of different socio-technical systems with varied foci, including sustainability certification, contract farming, consolidation of production, and improved use of inputs, aside from the development of sustainable technologies. In addition, 1M5R was promoted with incentives generated by a World Bank project and other initiatives in line with a national policy of increasing the quality of rice production for national and international markets. The interconnections of varied socio-technical systems, enacted by different intermediaries, catalyzed the spread of 1M5R. The widespread adoption by smallholder farmers increased their profits and raised awareness across diverse stakeholder groups of the higher marketability of rice produced with sustainable practices.

**Keywords:** irrigated rice; best management practices; sustainable production; scaling; Vietnam

## 1. Introduction

Rice is the staple crop in most Southeast Asian countries [1]. Recent analyses of yield gaps for lowland rice production identified opportunities to positively address yield shortfalls through the adoption of best management practices [2]. Subsequent trials of such best practices in the fields of farmers in Thailand [3] and Vietnam [4] showed increased production while reducing the amount of water, fertilizers, and pesticides. Further studies in Vietnam also have reported a significant reduction in greenhouse gases when best management practices for irrigated rice are implemented (Nguyen-Van-Hung et al., unpublished data). There are also increases in rice yield when best water management practices are followed [5]. Enabling the adoption of these best management practices is a crucial next step.

Vietnam is a major exporter of rice, with the Mekong River Delta (MRD) producing approximately 90% of the rice that is exported [6]. Farmers in the MRD typically have less than 5 ha for rice production, but they overuse chemical fertilizers and pesticides, which can have detrimental effects on water quality, biodiversity, and human health [3,7,8]. Various initiatives in the country aimed to address these problems through the introduction of sustainable rice production technologies. More recently, a national program promoted best management practices through "1 Must Do and 5 Reductions (1M5R)". It is essential to understand the outcomes of this initiative and trace the plausible pathways to capture the learning related to sustainability transitions. This review examines the development and subsequent wide-scale adoption of best management practices known as 1M5R in Vietnam. Veering away from equating the process of scaling with knowledge dissemination, we focus on the process of socio-technical change, which multiple actors have implemented. We do this through a review of secondary materials, comprising published papers as well as reports and presentations. These are complemented by interviews with actors directly involved in the development and scaling of 1M5R. The interview respondents were from varied stakeholder groups: researcher, policymaker, extension intermediary, and the private sector. Documenting this process contributes the a better understanding of sustainability transitions.

*Reconfiguration of Socio-Technical Systems*

We approach this review with the premise that the widespread uptake of any technology entails a complex reconfiguration of social and technical components [9,10]. There is a historical context that framed how the technology was developed. Furthermore, the technology changed over time through an adaptive process, with interactions between researchers, users (primarily farmers), and other stakeholders. The interactions considered the material and ecological aspects of reshaping the technology. We argue that the 'packaging' of 1M5R is a combination of historical influences, adaptive actions from researchers, farmers, and other stakeholders, and the communication campaign component that was tried and tested in Vietnam on a precursor set of best practices known as 3 Reductions, 3 Gains [11,12].

Scaling entails a reconfiguration emerging from interactions between biophysical, social, economic, and institutional changes in a socio-technical system [13]. Different actors, institutional arrangements, rules, and technologies comprise this socio-technical system. Intermediary actors are crucial and play roles in knowledge provision and translation, technology development, and public and industry processes [14–16]. There are systemic intermediaries who operate "in networks instead of 'one to one' mediation" [17]. They help to articulate the options and demand for knowledge, technologies or other processes, support the alignment of actors, and enable learning [18]. They also transform governance towards a transition process that can embed specific innovations in a system [19]. Linked to this are intermediaries that support the translation of new tools, techniques and processes within their networks [20]. These create new processes and incentive mechanisms. An example of this is contract farming. Mediated by the private sector, such contracts promote specific practices and provide incentives for these practices. In some cases, these intermediaries can be seen as cultural enablers because they re-create the social and organizational conditions that support an innovation. Lastly, there are also intermediaries that are focused on a niche or specialized product, service or market [21]. They connect experimental findings, or new ideas and products with the rest of the innovation system actors to enable changes to the current practices of varied actors. This is needed, as large-scale rice farming requires collective actions and alignment across different stakeholders in the value chain [22].

We hypothesize that varied actors (individual or organizational champions who acted as intermediaries) were instrumental in shaping the uptake of 1M5R in Vietnam. Some worked on the niche of 1M5R to develop the technology, test it with farmers, and show evidence of the benefits. There were also intermediaries that made connections in the socio-technical aspect, facilitating learning to fit the tools and techniques with the routines

of farmers and groups, as well as linkages between farmers and other stakeholders. Lastly, there were also systemic intermediaries that created legitimacy and provided broader incentive mechanisms that supported the socio-technical change. These intermediaries engage a socio-technical system wherein specific actors (e.g., farmers, service providers, contractors) employ different technologies or combinations of technologies [23]. They are linked together through specific organizational arrangements, rules, and incentives that inform their behavior in relation to 1M5R.

Within this theoretical lens, we trace the process of the technical development of 1M5R towards its uptake at the national policy level. We then examine how these policies were translated into government-supported programs, as well as the initiatives from other actors, such as those in the private sector. We pay particular attention to the Vietnam Sustainable Agricultural Transformation (VnSAT) Project as a major national program that explicitly scaled out 1M5R. Lastly, we review the outcomes in terms of adoption by farmers. As such, we examine the technological adaptation, organizational arrangements, intermediaries, incentives, and rules in the socio-technical system around 1M5R.

## 2. The Development of 1 Must Do 5 Reductions

### 2.1. Building on an Existing Framework

In 2003, the "Three Reductions, Three Gains" (Ba Giam, Ba Tang in Vietnamese) campaign was launched to reduce the use of seeds, fertilizers and pesticides. The use of mass media led to significant reductions in the use of these three inputs in the MRD [12]. The proponents of this campaign built their implementation by distilling sound scientific evidence into a simple heuristic, testing the Ba Giam, Ba Tang message through farmer participatory research [11]. A message design workshop then was used to develop materials for billboards, posters, and leaflets. This led to the development of a mass media approach to scale out their messages [24]. The local champions of Ba Giam, Ba Tang promoted the technology at a national level. A national committee was formed that reported to a vice minister of the Ministry of Agriculture and Rural Development (MARD). In 2006, the Minister of MARD proclaimed Ba Giam, Ba Tang as a national priority to be promoted by the National Agricultural Extension Center. The proponents of Ba Giam, Ba Tang claimed that they "reached" 2 million farmers [25], with an estimated benefit of US$44 per ha for farmers who adopted this approach [26].

In parallel with these activities, there was innovative field research on water management [5,27], ecologically based rodent management [28], and improved post-harvest management [29] of rice. There was also a realization that many farmers were not using good-quality seed. This led to an initiative to extend the success of Ba Giam, Ba Tang to embrace a broader set of best management practices for irrigated rice production. A committee consisting of international experts from the International Rice Research Institute (IRRI) and national experts was formed, with two initial tasks. One was to capture the best management practices in the written guidelines that initially targeted extension specialists. The second was to develop a simple heuristic to aid the promotion of the extended set of best practices. This gave birth to "1 Must Do, 5 Reductions" (Mot Phai, Nam Giam in Vietnamese). The "1 Must do" regards the use of certified rice seed. The "5 Reductions" pertain to seed, water, fertilizer and pesticide use, and reductions in post-harvest losses. The national champions of 1M5R had access to the national committee, which reported to the vice minister of MARD on the best management practices for rice production.

### 2.2. Adaptive Research Model for 1 Must Do, 5 Reductions

The development of 1M5R was based on evidence-based scientific findings obtained from:

(i) Baseline studies providing a needs assessment of smallholder farmers, via a rapid rural appraisal (RRA, detailed in Chambers [30]), coupled with national policy priorities (e.g., promotion of certified seeds).

(ii) Research trials of replicated plots conducted in the fields of farmers (scientist-led). At the end of each cropping season, the researchers met with farmers to discuss the findings and to plan, with input from farmers, what should be tested in the next season. The researchers had to adapt to the requests of the farmers, given that the trials were in the fields of farmers.

(iii) Engagement of farmers with the outcomes of the field trials and then requests for volunteers (early adopters) to test the new practices that appealed to them in a section of their rice fields. The farmers led these trials while researchers provided technical advice, certified seeds, and fertilizer. The latter was provided on the condition that it was applied following the recommended practice. The researchers assisted farmers in keeping a diary of their activities during the cropping season, and, at harvest, researchers collected data on yield and pest losses. Again, farmers and researchers met at the end of the season to discuss the findings and plan for the next season.

Local extension specialists arranged field days, which were attended by neighboring farmers, other extension staff, and researchers. Field visits to the adaptive research trials were also arranged for farmers from other districts in the province and extension specialists from other districts and provinces. The adaptive research approach we adopted is described further in Horne and Stür [31] and Flor et al. [32]. A scheme of the adaptive research model that was implemented in the MRD is shown in Figure 1.

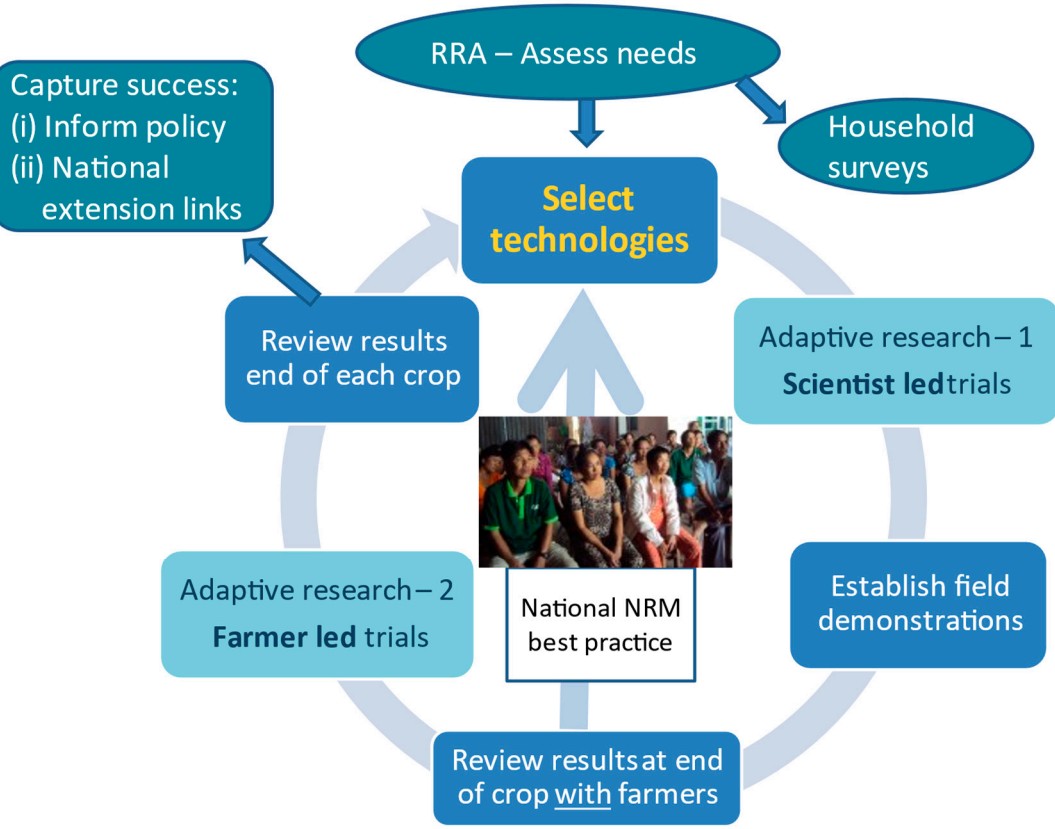

**Figure 1.** An outline of the adaptive research approach for developing evidence-based findings from trials conducted in the fields of farmers in the Mekong River Delta in Vietnam. (RRA = rapid rural appraisal; NRM = natural resource management).

Technological Adaptation

Following the development and promotion of 1M5R, several studies indicated that the use of inputs such as seed, pesticides, and fertilizers were still too high, with concerns regarding about the economic and environmental sustainability of rice production in the MRD [4,26,33]. Thus, following the adaptive research model described above, MARD and IRRI conducted a further testing, adapting, and refining of 1M5R in two pilot sites

in Can Tho city with the aim of optimizing rice productivity and reducing the negative environmental impacts of rice production in the MRD [4]. Simultaneously, research was conducted to evaluate the influence of a Large Rice Fields model and Good Agricultural Practice (GAP) on the management practices of rice farmers in the same locations.

The tested adaptations included submerging seeds in sodium chloride (NaCl) solution to remove potentially damaged and diseased seeds; sowing certified seeds using a manually pulled drum seeder to reduce seed rates; limiting pesticide use to two formulated product applications per pesticide group per season; installing field water tubes to determine when to irrigate the fields based on 'alternate wetting and drying' (AWD) recommendations; providing locally recommended guidance on fertilizer rates and timing; providing clear instructions to combine harvester operators to clean the harvester regularly to avoid spillage of grain, and to thresh rice slowly at the correct drum speed to minimize losses. The resulting combination of technologies included in the 1M5R is presented in Table 1. Some components of 1M5R have similarities with the components of Three Reductions, Three Gains; they specifically reduced seed rates, as well as leading to lower fertilizer and pesticide use. The adaptive research method is also an effective tool for promoting good practices fitted to a specific context, e.g., cropping systems and mechanization availability. For instance, the updated 1M5R standards identified the adoption of a target seed rate of 100 kg ha$^{-1}$ as the main challenge for farmers, given that farmers previously used seed rates in a typical range of from 175 to 230 kg ha$^{-1}$ in their conventional practice [4,5].

**Table 1.** Specifications of 1M5R applied in the field trial at Trung-Thanh Village, Co Do District, Can Tho.

| Criteria | Requirements | |
|---|---|---|
| Seed Rate * | $\leq$120 kg ha$^{-1}$ | |
| Seed Quality | Certified seed | |
| Nitrogen * | $\leq$100 kg ha$^{-1}$ | Applied with at least 3 splits |
| Insecticides * | Maximum 1 product application | No application within 40 days after sowing |
| Fungicides * | Maximum 2 product applications | No application after the flowering crop stage, except for the pre-harvest interval mentioned in label |
| Water Management | Alternate Wetting and Drying (AWD) | At least one mid-season draining following safe AWD practice |
| Harvesting | Using combine harvester | Harvest, when 80–85% of the grains per panicle are straw or yellow-colored |

* Overlap components with 3R3G (reduce seed rate, and fertilizer and pesticide use).

To promote the further scaling out of 1M5R, field demonstrations of these best management practices with farmer training were conducted across the region by MARD. Subsequent field trials with IRRI researchers involved an evaluation of the options to improve the efficiency of fertilizer application and other 'emerging' crop establishment methods to reduce seed rates, including mechanical transplanting (Nguyen-Van-Hung et al., unpublished data). In addition, the results from the field trials provided an evidence-based approach for field manuals and training materials, as well as for standards (e.g., limits in pesticide use) and key performance indicators that were established for farmers' organizations participating in the VnSAT project.

*2.3. Communication Strategy: Guidelines, Message Design, Ground-Truthing with Farmers*

A manual was developed, which provided guidelines for the implementation of each of the six practices promoted under 1M5R. The writing team consisted of national and regional representatives, supported by natural resource management and social scientists from IRRI. In the final workshop on the manual, there were English and Vietnamese versions, shown concurrently on different screens, so that changes could be discussed and agreed upon in a reasonable timeframe. The manual was completed in May 2009,

together with a message design workshop to develop materials to promote 1M5R. For an interactive message design, facilitators created three large billboard designs. An outline of the manual and the three options for the billboard/banners were then discussed with a local farmer group. The farmer group provided essential input. For example, the design that they liked was of a woman holding freshly harvested rice, with some mountains in the background. Their feedback was that a "pretty" lady who was not wearing a traditional hat would be ignored by most farmers, because they would relate the image with advertising by a chemical company. Additionally, there are very few hills in the MRD and certainly no mountains, so they requested that they be removed. The final billboard is shown in Figure 2.

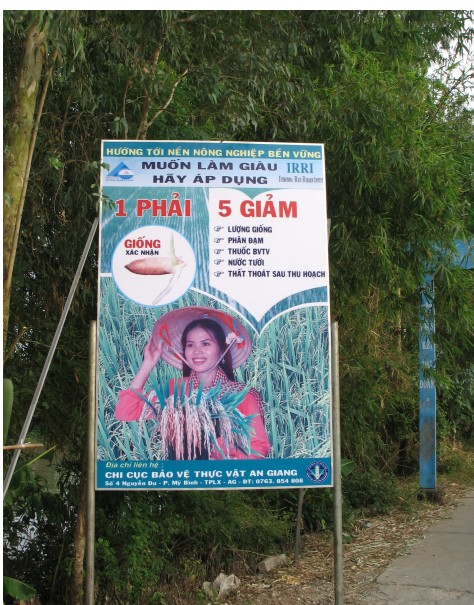

**Figure 2.** Billboard of 1 Must Do, 5 Reductions (Mot Phai, Nam Giam) along a road in a rice production area in An Giang province in the Mekong River Delta.

### 2.4. High Profile Roll-Out of Best Practices for Rice Production under 1 Must Do, 5 Reductions

The manual and promotional material (billboards, flags, leaflets) were finalized for production in May 2009. A high-profile roll-out of 1M5R in An Giang province did not occur until November 2009. The delay was necessary because the national committee needed to approve the materials and the planned campaign. Permission to proceed was achieved.

The roll-out of 1M5R in November 2009 occurred at the front of the People's Committee Building of An Giang province, with a 20 × 10 m banner promoting the event (Figure 3). The vice-chairman of the People's Committee, the deputy director of the National Plant Protection Center, the director of the provincial department of agriculture and rural development and three IRRI scientists, were among the people who attended the launch. Moreover, the private sector, with an interest in fertilizers and other inputs, was represented in the event [33]. There was strong media presence and, after the formal launch, 50 riders with 1 Phai 5 Giam banners set off in formation to tour the city and then tour three other provinces over the next two days. During the previous week, there were 20 billboards (Figure 2) erected in the province, plus 2000 posters displayed in public sites such as schools, coffee shops, farmer clubs, and hospitals. There was a print run of 17,000 brochures and 8000 manuals. 1M5R received an impressive launch!

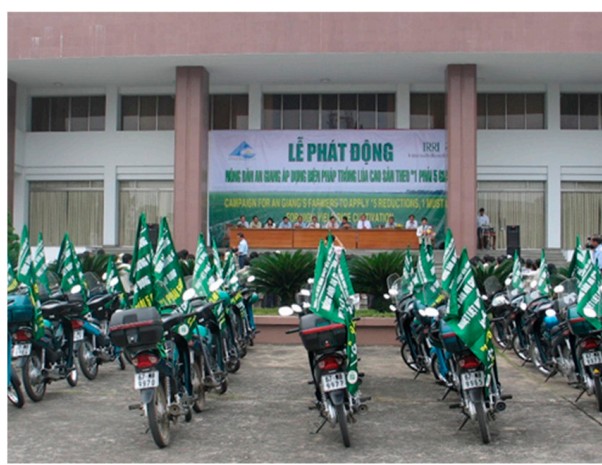 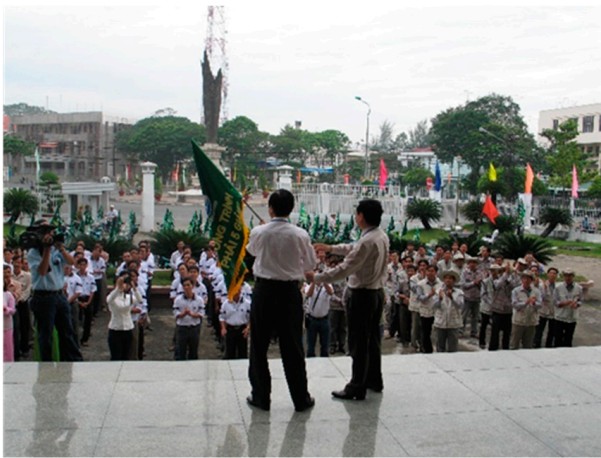

**Figure 3.** National launch of 1 Must Do, 5 Reductions (Mot Phai, Nam Giam) at Long Xuyen, An Giang, in November 2009.

On 7 November 2012, Mot Phai, Nam Giam was declared by the vice minister of MARD as a nationally approved best practice for irrigated rice production.

## 3. Pathways to Impact

### 3.1. Government Support

With the government endorsement of 1M5R, outreach was supported by an infrastructure of government organizations with mandates for governance and extension. Various local champions across this network of government agencies provided policy support and legitimacy for the promotion of 1M5R. While the MARD provided the overarching policy direction at the national level, the Department of Agriculture and Rural Development (DARD) provided a central hub for 1M5R extension and governance, as well as regulatory control for imports and plant quarantine [34]. At a provincial level, the Sub-Department of Crop Production and Plant Protection (Sub-DCPPP) created a mandate to manage crop varieties and production, fertilizers, use of agricultural land, plant protection, plant quarantine, pesticides, relevant extension activities, and international cooperation. The Sub-DCPPP implements these under DARD and is well-placed to support the extension of sustainable management technologies. According to interviews with staff from Sub-DCPPP, they were responsible for directly building the local capacity to promote the adoption of 1M5R within the provincial area. They also have an advisory role within the provincial DARD, and interact with researchers, universities, institutions, and private sector partners in promoting the use of 1M5R. More importantly, Sub-DCPPP coordinated activities related to 1M5R at district and commune levels.

In the villages, the Sub-DCPPP staff that were interviewed said they supported groups of farmers who are members of agricultural cooperatives established at a commune level. They implemented hands-on training using farmer field schools, established field demonstrations and on-farm workshops. Arrangements to promote 1M5R helped farmers understand the benefits and encouraged them to implement demonstration trials. The Sub-DCPPP also engaged in the preparation of extension materials for distribution to farmers, and the promotion of 1M5R through public media such as newspapers, radio and television. In different provinces, the Sub-DCPPP developed their own programs to engage farmers in adopting 1M5R. For example, in Can Tho, in addition to the regular mandated activities, various contests were held for farmer groups to facilitate group learning and promote the adoption of 1M5R. the interviewed staff said the groups competed based on 1M5R practices. The prize for winning cooperatives was small piece of agricultural equipment such as seed sprayers.

Aside from their extension activities, the Sub-DCPPP supports and encourages farmers to form agricultural cooperatives or join existing ones in their communes. They also facilitate linkages between farmers and private sector groups. The latter engage farmers

in production activities that enable companies to improve the marketing and selling of their produce. This can occur through contract farming or through linkages for inputs and the coordination of production in groups. The Sub-DCPPP also plays a regulatory role and monitors technology adoption by farmers, particularly those who have been trained on 1M5R.

### 3.2. Small Farmers, Large Fields (SFLF)

Complementary policies and initiatives from the government were implemented alongside the introduction of 1M5R to farmers. One initiative was kickstarted by MARD in 2013, following the Prime Minister's Decision to facilitate cooperative development, linkages between rice production and market, and the consolidation of production through a Large Rice Fields model (LRF) [35]. The policy, passed at the central level, is focused on the Mekong Delta region of Vietnam 62/2013/QĐ-TTg, 25 October 2013). Although officially termed LRF, a common name for the program is Small Farmers, Large Fields (SFLF).

The LRF aims to support the development of associations for the coordinated production and marketing. This policy supported the formation of farmer groups into cooperatives that coordinate their production and bulk out their produce for enhanced linkage to markets. Groups are formed of about 300–500 hectares of adjacent farms [36]. In strengthening these groups, the farmers were taught and then required to practice 1M5R. Observing the implementation of this, the staff of the Plant Protection Department (under MARD) noted that "the unification of production also enabled farmers to implement 1M5R".

### 3.3. Good Agricultural Practice (GAP) Certification

Another parallel initiative was the establishment of sustainability and certification standards for rice production through the Vietnamese Good Agricultural Practice (Viet-GAP) program. MARD issued a policy in 2010 supporting VietGAP (2998/QĐ-BNN-TT, 9 November 2010). With the proliferation of contract farming and the consolidation of farmers into cooperatives, the next step was to develop a system that could allow farmers to obtain a price premium for sustainably produced rice. The VietGAP program linked contract companies with farmer cooperatives to provide price incentives to those that became certified [36]. The VietGAP built upon the technological components and training of 1M5R, as well as the consolidation of farmers into cooperatives under LRF [36].

### 3.4. Vietnam Sustainable Agricultural Transformation Project

In 2015, the World Bank supported a large-scale program of Sustainable Agricultural Transformation (VnSAT), which was implemented by MARD. The project covered two major commodities, including rice in the MRD and coffee in Central Highlands. For rice, two components are directly relevant: institutional strengthening to support agricultural transformation and support of sustainable rice-based systems [37]. This entailed the provision of technical and financial support for two groups of actors in the rice value chain: rice farmers and millers/processors.

For rice farmers, VnSAT supported technical training and field demonstrations for farmer organizations. VnSAT also provided grants to support farmer organizations in the multiplication of certified seeds, investing in the reduction in post-harvest losses, and improving their marketing. Small-scale infrastructure improvements, such as inter-field road, inter-field channels, better access to electricity connections, co-investment in pumps, and improved irrigation, were provided based on the group's needs. Linkages between groups and contractors were facilitated to improve production and provide incentives for sustainable practices. These linkages were facilitated by technical support unit in the eight provincial DARDs in the MRD.

For the private sector actors, the project invested in upgrading rice-processing technology and facilities through competitive medium- and long-term loans (4 to 7 years) via selected commercial banks. The project also provided support to the technical departments and relevant agencies of MARD and the eight provincial DARDs to improve their agri-

cultural extension capacity, increase the production of foundation seeds, improve seed certification, and monitor GHG emissions.

The MARD incorporated its national strategies into VnSAT priorities and targets, so that 1M5R was intricately embedded into the project. According to an implementer of the project, 1M5R recommendations are the technical components of the training of rice farmers in the best production practices under VnSAT. Environmental outcome targets such as lower GHG measurements and reduced input use are among the performance indicators of the project. These targets were set in line with the expected benefits of the adoption of 1M5R by farmers.

This project's major support to the scaling of 1M5R was through the technology demonstrations coordinated in a wide geographic scale by researchers, extension staff, cooperatives, and the private sector ([38], Hung et al. unpublished data). After training, farmers' cooperatives were encouraged to trial the 1M5R in gradually expanding areas until they reached the set adoption target. According to the implementers, once this adoption target was reached, the cooperative qualified for a project investment of up to USD400,000. The farmers' cooperative then had to develop a business case to operate and benefit from the facilities and infrastructure to receive the funds. Examples of proposed investments include the installation of a water pump, repair or improvement of an irrigation canal, or purchase of agricultural machinery. This benefit is for the whole group, with each cooperative being composed of 500 rice farming households or cultivating an area of around 500 ha. Thus, the VnSAT project generated incentives for cooperatives and individual farmers to adopt 1M5R.

VnSAT provided financing through local banks to enable private sector investments in machinery and processing. Millers could, for example, obtain loans to upgrade their facilities. This enables the processing side to handle the bulk of the produce being collected from the cooperatives in the area.

Adult Learning Strategy to Increase 1M5R Uptake

The multi-pronged VN-SAT intervention was accompanied by an adult learning strategy that catalyzed a social change process to shift farmers' beliefs, attitudes, and understanding of their farming system and social networks. Through this learning, farmer groups bolstered the practice of 1M5R by individual farmers. Actors in the rice value chain were also targeted, such as farm laborers, harvesting service providers, millers, and rice traders, so that they supported the shift in the behavior of farmers.

As Tuan and Singleton [39] described, the aim of this strategy was to balance the top-down extension approach that is still common in various areas. Hence, it was predicated that the groups of farmers examine their own challenges and brainstorm relevant opportunities (Figure 4). The experience gained with VnSAT indicated that farmers are not simply knowledge-receivers and/or -adopters. They screen new knowledge and require facilitated learning with other stakeholders through a well-planned process of change. An entry point for this process was to engage local extension specialists to facilitate constraint analyses. These analyses targeted large geographical areas (such as province or district) as well as more site-specific analyses for smaller areas (such as commune or village). Further along the learning process, farmers were able to understand the new technologies, to experiment, and to implement those technologies in a manner suited to their farming conditions. Hence, technology demonstrations at various VnSAT sites became learning tools to understand and build trust in the technology.

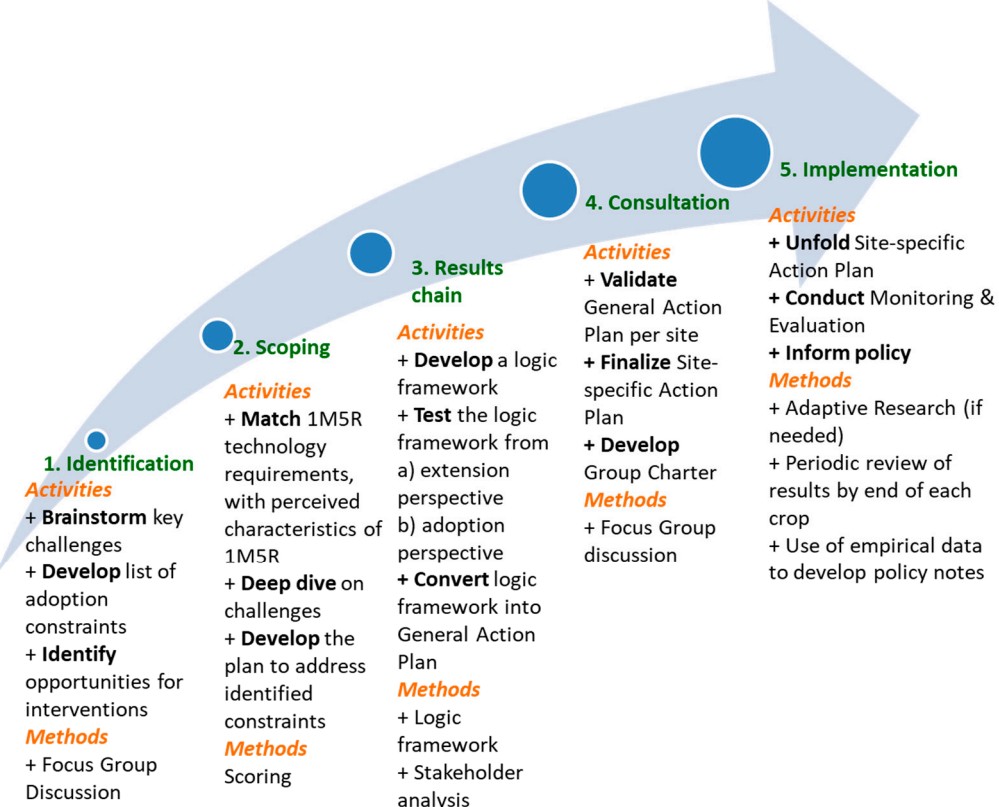

**Figure 4.** Five steps of the Adult Learning Process under Vietnam Sustainable Agricultural Transformation project (VnSAT) with the activities and methods for each. Adapted from Tuan and Singleton [39].

More specifically, the strategy provided a 5-step procedure to guide learning activities. The stepwise procedure is as follows: Identification, Scoping, Results Chain, Consultation, and Implementation (Figure 4). The first three steps involved local extension staff, and the last two steps engaged farmers to analyze and address site-specific issues to promote solutions and adoption (Figure 4). By participating in these five steps, an action plan was produced to enable both local extension workers and farmers to agree on what actions were to be taken, by whom, and how these actions were to be monitored or adjusted during the adoption process. The action plan embraced the five stages of adult learning and was instrumental in assisting the farmers in implementing and achieving the sustained adoption of 1M5R.

### 3.5. Private Sector Engagement on 1 Must Do, 5 Reductions

The primary interests of Vietnamese rice traders and exporters on sustainable production are to limit pesticide residues in rice, improve the quality and brand of Vietnamese rice in the international market, and raise the profits of farmers [40]. Thus, companies such as the Loc Troi Group have adopted 1M5R and invested in training farmers on the associated best practices [41]. Loc Troi establishes contracts with cooperatives of farmers who are willing to achieve certification for their rice production at an agreed buying price of the produce. The certification is received through the Sustainable Rice Platform (SRP) standards, but the group uses 1M5R as the basic set of practices that enable farmers to pass certification standards. According to a representative, the group has a significant reach, with about 500 extension staff. The company engages up to 90,000 contract farmers in the main rice season. Loc Troi is structured in four sectors, with specific roles: plant protection, agricultural services, food (management and trade), and the seed sector. These sectors work in a coordinated manner to supply inputs and services to the contracted cooperatives, and then collect the produce for trade.

Through their extension staff, a company representative further described that Loc Troi provided knowledge on 1M5R practices. The staff ensure that the inputs provided are measured in advance to meet the requirements for SRP certification. They guided farmers to implement farm activities and document their practices for certification. Then, the farmer group leaders and cooperative officials monitored the farmers. Thus, the adoption of 1M5R practices was embedded in the routines that Loc Troi established in its contract farming. Farmers are interested in contract farming because of the potential for and observed higher prices for their rice, as well as the secure investment from inputs and the agricultural machinery services from the company.

Another private sector initiative is from the input companies. The PPD linked established farmer cooperatives with input companies, specifically for fertilizers. These companies provide inputs to farmers, and they were trained to produce rice following the 1M5R guidelines. This created a market for the companies and a secured source of input for farmers. A concrete example of this is the Smart Rice Farming (SRF) initiative, funded by the Binh Dien Fertilizer Joint Stock Company and implemented by the National Agricultural Extension Center. According to one government staff member involved in the implementation, the SRF worked with farmer groups on 5–10 hectares and supported them with training, tools, and inputs (fertilizers). The group followed recommendations aligned with 1M5R.

## 4. Evidence of Uptake at the Farmer Level

After five years of the promotion of 1M5R through the VnSAT project, 1M5R was adopted in approximately 114 thousand ha in the MRD (Table 2). In some areas, 3R3G was introduced first, and these practices were adopted in higher numbers. Notably, 3R3G has some of the components of 1M5R. More importantly, of the farmers who were part of VnSAT and used either 3R3G or 1M5R, there is evidence of higher incomes, of between 19% to 36%, across different provinces (Table 2).

**Table 2.** Number of farmers trained, area covered, and area of adoption of Three Reductions, 3 Gains (3R3G) and 1 Must Do, 5 Reductions (1M5R), with percentage of increase in income comparing farmers within the Vietnam Sustainable Agricultural Transformation project (VnSAT) and those not in the program (non VnSAT).

| Province. | Three Reductions, Three Gains | | | One Must Do, Five Reductions | | | Percentage Increase in Income (VnSAT Farmers vs. Non VnSAT Farmers) |
|---|---|---|---|---|---|---|---|
| | Number of Farmers Trained | Area Covered in Training (ha) | Area with Farmers Adopting 3R3G (ha) | Number of Farmers Trained | Area Covered in Training (ha) | Area with Farmers Adopting 1M5R (ha) | |
| An Giang | 18,833 | 25,417 | 20,969 | 18,833 | 25,417 | 19,851 | 26.4 |
| Cần Thơ | 25,435 | 31,794 | 26,997 | 17,805 | 22.256 | 17,263 | 34.6 |
| Đồng Tháp | 18,624 | 31,657 | 24,232 | 8,829 | 15,207 | 11,541 | 27.5 |
| Hậu Giang | 21,645 | 21,976 | 18,677 | 14,278 | 14,680 | 13,586 | 25.8 |
| Kiên giang | 19,240 | 31,927 | 24,394 | 9,654 | 16,917 | 11,165 | 30.3 |
| Long An | 11,835 | 29,440 | 22,220 | 10,866 | 28,606 | 18,653 | 29.0 |
| Sóc Trăng | 16,746 | 22,329 | 20,106 | 13,351 | 15,598 | 13,712 | 18.9 |
| Tiền Giang | 23,422 | 19,123 | 17,847 | 10,832 | 8201 | 8099 | 36.4 |
| Total | 155,780 | 213,663 | 175,442 | 104,448 | 124,648 | 113,870 | |

**Source:** VnSAT Workshop on implementation progress, monitoring and evaluation on 26 May 2018, Can Tho City and 15 December 2021, Hanoi, Vietnam.

By 2020, the number of farmer organizations that were reached by VnSAT and that are adopting 1M5R practices have increased (Table 3). In 2018, the total area under contract farming and implementing sustainable practices was 18,000 ha. By 2020, this increased to more than 63,000 ha (Table 3).

**Table 3.** Number of farmer organizations reached and those who engaged with contract farming with area coverage in the Mekong River Delta, 2018 and 2020.

| | | 2018 | | | 2020 |
|---|---|---|---|---|---|
| Province | Total Farmers Organization | Farmers Organization with Contract Farming | Percentage with Contract Farming | Area Adopting Sustainable Practice with Contract Farming | Area Adopting Sustainable Practice with Contract Farming |
| An Giang | 40 | 27 | 67.5 | 3848 | 3842 |
| Can Tho | 21 | 15 | 71.4 | 2536 | 12959 |
| Dong Thap | 45 | 28 | 62.2 | 2825 | 4432 |
| Hau Giang | 36 | 32 | 88.9 | 613 | 8800 |
| Kien Giang | 54 | 38 | 70.4 | 4505 | 10,231 |
| Long An | 19 | 8 | 42.1 | 1097 | 6848 |
| Soc Trang | 25 | 12 | 48.0 | 1293 | 7894 |
| Tien Giang | 20 | 9 | 45.0 | 1269 | 8389 |
| Total | 260 | 169 | | 17,986 | 63,395 |

**Source:** VnSAT Project Workshop on implementation progress, monitoring and on 26 May 2018, Can Tho City and 15 December 2021, Hanoi, Vietnam.

Furthermore, independent studies found a high rate of adoption among farmers. Connor et al. [42] implemented a study on the adoption of 1M5R by farmers in An Giang and Can Tho provinces. A cross-sectional survey among project farmers ($n$ = 465) revealed that farmers followed the requirements specified under 1M5R. Most farmers followed the requirements of pesticide reduction ($n$ = 346, 74.4%), fertilizer reduction ($n$ = 343, 73.8%), post-harvest loss reduction ($n$ = 463, 99.6%), and the use of certified seeds ($n$ = 421, 90.5%). The majority of the farmers used two to three of the recommended 1M5R technologies ($n$ = 367, 78.9%), and the most-adopted technologies were combine-harvesters ($n$ = 463, 99.6%) followed by AWD ($n$ = 161, 34.6%) and high-yielding varieties ($n$ = 131, 28.2%). Most farmers ($n$ = 399, 85.8%) reported that they reduced their seed rate. Indeed, farmers reduced their seed rate by 77 kg ha$^{-1}$ (SD = 40 kg ha$^{-1}$), which accounts for an estimated 30% decrease. However, they still used an average of 164 kg ha$^{-1}$ (SD = 36 kg ha$^{-1}$) rate, compared to the recommended 100–120 kg ha$^{-1}$.

Twenty-two (4.7%) farmers reduced their seed their seed rate quoted several constraints, including that the weather conditions did not allow for a reduction in seeds, that they expected low yield and that it was too difficult to apply.

In addition, almost half of the surveyed farmers reduced their water use ($n$ = 211, 45.5%). Farmers mentioned that this practice was difficult, that weather conditions did not favor the reduction in water use and that it does not fit their cropping patterns. This shows that farmers face external physical barriers, as well as internal personal factors. This is particularly highlighted in the results evaluating the drivers of adoption.

The main drivers for adopting the whole 1M5R package were the ease of implementation of the single requirements, farmers' educational attainment, their satisfaction with the whole program, and their non-rice income. Farmers, in general, perceived a variety of benefits of adopting the 1M5R requirements, including the fact that most of them were easy to apply, that they had lower labor costs, that the technology or practice was less expensive, and that they fitted their cropping pattern [42].

In another study, Wehmeyer (unpublished data) conducted a comparative analysis from survey data collected in 2014 and 2019 to ascertain how many farmers were reached by the projects (treatment) and those who were not (control). The results of the 2019 survey showed that project farmers were introduced to a variety of technologies and practices, which helped to achieve the goals of 1M5R. All project farmers adopted at least one technology, including technologies that are complementary to 1M5R. The adoption rate was particularly high for the combine harvester (100%), drum seeder (95.4%), alternate wetting and drying (93.7%), and the use of improved varieties (88.8%). Considerable spillover effects occurred over the five-year period; control farmers also adopted the combine harvester (100%), drum seeder (75.8%), alternate wetting and drying (78.1%), and

the use of improved varieties (92.2%). In general, farmers indicated that the benefits of adopting one or more 1M5R technologies were reduced labor (95.6%), lower input cost (79.3%), better yield (65.9%), and better crop stand (57.8%).

## 5. Discussion and Conclusions

The widespread adoption of 1M5R as a foundation for the promotion of sustainable rice production in the Mekong River Delta is an impressive success story. The success not only strengthened the economic status of smallholder farmers but also provided a strong platform for promoting sustainably produced rice for both national and international markets. At the national level alone, there is compelling evidence that people in larger cities in southern Vietnam are prepared to pay a premium for what they perceive to be healthier rice [43].

The scaling of 1M5R in the Mekong River delta of Vietnam that is examined in this review showed a series of events including incentives through policy change, adaptive actions, and the creation of a supportive context for adoption, including the incentives provided through the VnSAT project. The process of socio-technical change that embedded 1M5R practices into the routines of rice farmers was affected by multiple actors in a complex process, similar to that theoretically discussed by Wigboldus et al. [13]. This was built on the work of a niche of stakeholders who implemented adaptive research and provided a basis for technical and policy recommendations. With this evidence, strategic communication and policy backing were used to introduce and promote 1M5R.

We initially expected that the scaling process involved one socio-technical system focused on 1M5R, which, in turn, was reshaped by the diverse range of actors and interest groups involved. In this review, we found a convergence of varied socio-technical systems, each with a different focus, which has promoted 1M5R in various ways. We illustrate this in Figure 5.

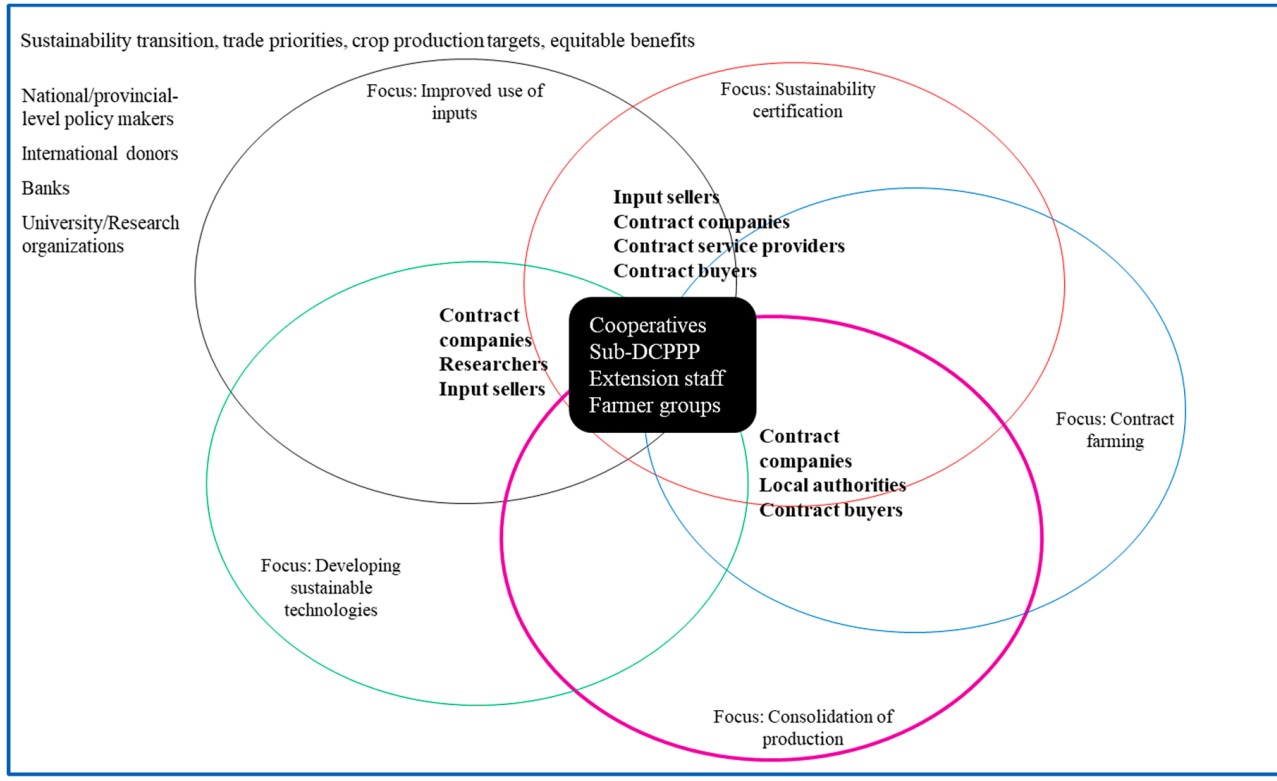

**Figure 5.** Convergence of socio-technical systems, each focused on a specific shared interest, with linked intermediaries and set in a broader context that supported the uptake of 1 Must Do, 5 Reductions (1M5R) in the Mekong Delta region of Vietnam.

We find, for example, that the initiatives regarding a sustainability certification, whether for the standards established under VietGAP or SRP, linked different intermediaries and created new institutional arrangements. These are not specific to the promotion 1M5R, because the standards entail other technologies and practices, beyond 1M5R. The intermediaries, however, used 1M5R as a technology base and employed certification and infrastructure support as incentive mechanisms for the adoption of 1M5R. This is a separate socio-technical system, with its own specific focus, incentives, and technologies (Figure 5). Vellema and van Wijk [44] studied sustainability standards and certification in global food chains. They found that multi-stakeholder partnerships enable the co-creation of standards to fit local norms, rules, and practices. In applying these, the group of actors is (re)shaping the socio-technical system.

Another socio-technical system, around the consolidation of production through an LRF model (Figure 5), is focused on strengthening these groups' voice when engaging with traders. Ba et al. [43] found that the LRF model promoted horizontal coordination among farmers. Their study also showed there several types of contract arrangements were made, and the vertical integration between farmers and exporters in the Mekong Delta of Vietnam facilitated these arrangements. The focus of this socio-technical system was not a technology; however, in the training for consolidated production, 1M5R was promoted to farmers. A study on the adoption of aquaculture technologies in Vietnam also showed why the clustering of producers resulted in higher levels of adoption of sustainable production technologies. Joffre et al. [45] found that the frequency of interaction and trust among farmers, as well as between the farmers and value chain stakeholders (vertical integration), influenced their adoption decisions.

The initiatives around contract farming form yet another socio-technical system with specific organizational arrangements, incentives, and rules (Figure 5). These rules and arrangements are implemented by rice traders and input suppliers (often the same company in the Mekong River delta), together with other stakeholders. The focus is on providing inputs and services, and then buying the produce. In this system, sustainability may not always be the main interest, but there are intermediaries interested in buying sustainably produced rice. Considering this, they also promote 1M5R and integrate the practice in their contract arrangements. The convergence of these varied socio-technical systems, including knowledge extension, group formation, group learning, and reshaping interest and incentives, led to an alignment of the practices of farmers and 1M5R.

A notable part of this process is the bridging that is conducted by the different intermediaries. As shown in Figure 5, some intermediaries are common across different systems. They influence the processes happening in parallel. Such innovation brokering has been highlighted by others [46,47]. At the same time, these intermediaries do not always share the same focus, motivation, or interest. Thus, there are often conflicts, negotiations, balancing, and trade-offs at play. While some actors may be keen on reducing pesticides or fertilizers, for example, others may not be as interested. These complexities happen concurrently when farmers learn 1M5R. An aspect that requires more research is the transition processes that moderate the conflicting interests and challenges in aligning these different strategies, actors, and activities. It would also be of interest to see how other technologies approved at the national level could build upon 1M5R, and extend the technological options that are included.

This case emphasizes that scaling is not solely a process within the socio-technical system of the innovation in question, in this case, 1M5R. The interconnections of varied socio-technical systems are enacted by different intermediaries that catalyze the spread of 1M5R. Previous studies on scaling indicate that this convergence can shape frameworks and everyday practices, integrating the innovation [23,48]. Furthermore, the economic mechanisms and contextual factors underlying specific policies can make or break the momentum and lead to eventual success in scaling sustainable technologies [49,50]. In the Mekong River Delta, there were economic benefits to the farmers who adopted 1M5R, which maintained the momentum for the outreach of best practices for sustainable rice

production. While supportive policies may be present, an alignment is required between incentive mechanisms, institutional factors, knowledge, and markets, alongside the adapted technologies. The scaling of 1M5R was not a simple linear process from research to adoption, nor was it solely pushed by a top-down implementation of policies. These insights from 1M5R could be useful for other countries that have programs where a tested set of technologies is being promoted at a national level; for example, Integrated Crop Management in Indonesia. Understanding the process of this review provides insights for the scaling of other innovations in the agricultural sector and for sustainability transition.

**Author Contributions:** Conceptualization: G.R.S., R.J.F., L.A.T., N.V.H., B.O.S.; Data collection: N.V.H., N.T.M.P., M.C., A.M.S., H.W., R.J.F., L.A.T., B.T.C., H.T., G.R.S. All authors have read and agreed to the published version of the manuscript.

**Funding:** This work was implemented with the support of the World Bank through the Vietnam Sustainable Agricultural Transformation Project (VnSAT; Credit No. Cr. 5704-VN, Contract No.: SSS-02 from Central Project Management Unit of the Ministry of Agriculture and Rural Development), and the Swiss Agency for Development and Cooperation (SDC) through the Closing Rice Yield Gaps with Reduced Environmental Footprint (CORIGAP Pro) (Grant number 81016734).

**Acknowledgments:** The authors are grateful to Ho Van Chien, Pham Thi Minh Hieu, Tran Thi Kim Thuy, Pham Van Quynh, Cao Vinh Thong, Dang Thanh Phong, and Tran Nguyen Ha Trang for their valuable insights on the various initiatives in Vietnam. Photos in the article are from Grant Singleton. The International Rice Research Institute is a member of One CGIAR.

**Conflicts of Interest:** The authors state no conflict of interest.

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
