# Peer review of "Unpacking the Processes that Catalyzed the Adoption of Best Management Practices for Lowland Irrigated Rice in the Mekong Delta"

_agronomy, doi:10.3390/agronomy11091707_

Round 1
Reviewer 1 Report
The paper titled 'Unpacking the processes that catalyzed the adoption of best management practices for lowland irrigated rice in the Mekong Delta' is more a report than a review, based on an analysis of results of the M15R project. The authors put a lot of effort to investigate and describe the processes which took place during the implementation of new practices in rice production in Vietnam, that reduce the impact of agricultural production on the environment. The paper is a case of study, which can be helpful in the implementation of similar projects in other regions of the world, aiming at adaptation agriculture to predicting climate change. It is worth underlining, that implementation of sustainable practices in rice production resulted in appearing a convergence of varied socio-technical systems, each with a different focus that has promoted 1M5R in varied ways. This case and other studies recalled by the authors, on scaling indicate that such convergence can shape frameworks and everyday practices integrating the innovation. What is more, economic mechanisms and contextual factors underlying specific policies can make or break the momentum and eventual success in scaling sustainable technologies.
Please, the authors to unify the transcription of units (kg/ha or kg ha-1) (line 208, 209, 488, 489 and the content of Table 1).
I recommend this paper to be published in the journal Agronomy.
Author Response
Comment: The paper titled 'Unpacking the processes that catalyzed the adoption of best management practices for lowland irrigated rice in the Mekong Delta' is more a report than a review, based on an analysis of results of the M15R project. The authors put a lot of effort to investigate and describe the processes which took place during the implementation of new practices in rice production in Vietnam, that reduce the impact of agricultural production on the environment. The paper is a case of study, which can be helpful in the implementation of similar projects in other regions of the world, aiming at adaptation agriculture to predicting climate change. It is worth underlining, that implementation of sustainable practices in rice production resulted in appearing a convergence of varied socio-technical systems, each with a different focus that has promoted 1M5R in varied ways. This case and other studies recalled by the authors, on scaling indicate that such convergence can shape frameworks and everyday practices integrating the innovation. What is more, economic mechanisms and contextual factors underlying specific policies can make or break the momentum and eventual success in scaling sustainable technologies.
Please, the authors to unify the transcription of units (kg/ha or kg ha-1) (line 208, 209, 488, 489 and the content of Table 1).
Response:
We thank the reviewer for the review and comments. We have made the changes accordingly.
Reviewer 2 Report
The manuscript outlines diffusion of best practices for rice producers in Mekong Delta, with particular attention to socio-technical change and process scalability. Has this been a demonstrable testbed for comparable system adoption elsewhere?
Pesticide Action Network UK is not a reputable author affiliation for a peer-reviewed journal article (LINE 11)
Curious about the convergence and applicability of current and forthcoming technologies to 1M5R, such as hybrid rice, biotech rice (in the public domain), CRISPR, etc. and associated constraints. This the epitome of “socio-technical”.
Show interrelational/intersectional schematic of 3 Reductions, 3 Gains (as a precursor to 1M5R), and 1M5R to VnSAT in a Venn diagram? Can also include Large Rice Fields model and GAP (SECTION 2) Can additionally include SFLF in section 3.2 and VietGAP or SRP as practices “beyond 1M5R” (LINE 544) – this is much in the way the “beyond organic” mantra has taken a foothold in the US, etc.
What are the constituents of a Rapid Rural Appraisal? Include table? Or simply “as outlined by author(s) et al.” (LINE 160) Can word the same way that the adaptive research approach “is described further in Horne and Stur and Flor et al. (LINE 178)
Mention that IRRI is a constituent of CGIAR.
Table 1. Insecticides – no application within 40 days after sowing. Is this a label prescription or a best practice?
Section 2.3 – were cell phones not utilized for the promotional rollout? My understanding is that diffusion of cell phones is fairly robust, even in rural areas, and often used by Extensionists (at least in the Philippines, would assume Vietnam is similar).
LINE 257 Extra space before Moreover
LINE 459 “Notably, 3R3G has some of the components of 1M5R” Can commonalities be outlined in a table? OR include a summary timeline/chronology of the progenitor and successor systems (starting with TRTG and ending with “beyond” 1M5R) with associated years and core practices - would be a useful addition to observe the evolutionary continuum.
Author Response
- The manuscript outlines diffusion of best practices for rice producers in Mekong Delta, with particular attention to socio-technical change and process scalability. Has this been a demonstrable testbed for comparable system adoption elsewhere?
Mentioned similar national program in Indonesia, on Integrated Crop Management (Widjayanthi 2020). The scaling of ICM could benefit from the insights from scaling of 1M5R.
We added in the concluding section (Lines 606-609): “These insights from 1M5R could be useful for other countries that have programs where a tested set of technologies are being promoted at a national level; for example Integrated Crop Management in Indonesia.”
- Pesticide Action Network UK is not a reputable author affiliation for a peer-reviewed journal article (LINE 11)
The co-author was formerly with IRRI but has since moved to this organization. We cannot change this affiliation.
- Curious about the convergence and applicability of current and forthcoming technologies to 1M5R, such as hybrid rice, biotech rice (in the public domain), CRISPR, etc. and associated constraints. This the epitome of “socio-technical”.
With 1M5R, the technologies were tested and then approved by government institutions. This was a primary step in the scaling process. The technologies mentioned by the reviewer could be interesting to speculate, but as these were not included in the government program, we could not include them in the paper.
We concur with the line of thinking of the reviewer however, and added in the discussion section (Lines 588-520) the following:
“It would also be of interest to see how other technologies approved at national level could build upon 1M5R, and extend the technological options that are included.”
- Show interrelational/intersectional schematic of 3 Reductions, 3 Gains (as a precursor to 1M5R), and 1M5R to VnSAT in a Venn diagram? Can also include Large Rice Fields model and GAP (SECTION 2)Can additionally include SFLF in section 3.2 and VietGAP or SRP as practices “beyond 1M5R” (LINE 544) – this is much in the way the “beyond organic” mantra has taken a foothold in the US, etc.
In line 205-206, we clarified similarities in terms of technical recommendations.
“Some components of 1M5R have similarities with the components of Three Reductions, Three Gains, specifically reduced seed rates, as well as lower fertilizer and pesticide use.
We also specified these in the Table 1.
The other programs are based on different requirements. For example, GAP consists of the rules, and procedures that guide producers to produce, harvest, process and market agricultural commodities to meet a number of requirements. These include the requirements to do with food safety and quality, product traceability and environmental protection .The SRP standard consists of 41 requirements including production requirements, resource use efficiency, greenhouse gas emissions and social indicators such as labor rights worker safety and wages.
There is no direct overlap between 1M5R and SFLF, VietGap and SRP as the detailed specifications differ; however, we do acknowledge that farmers’ producing under 1M5R could also be meeting the SRP requirements on fertilizer and pesticide use as well as water management, they can be producing under GAP and SFLF meeting all the extra requirements. Therefore, it is not possible to depict an interrelation between all the programs mentioned but we highlight the similarities with 3R3G as a precursor for 1M5R.
- What are the constituents of a Rapid Rural Appraisal? Include table? Or simply “as outlined by author(s) et al.” (LINE 160)Can word the same way that the adaptive research approach “is described further in Horne and Stur and Flor et al. (LINE 178)
In Line 161, we added a reference to RRA detailed description from Chambers.
- Mention that IRRI is a constituent of CGIAR.
In the acknowledgements, we added: The International Rice Research Institute is a member of the One CGIAR.
- Table 1.Insecticides – no application within 40 days after sowing. Is this a label prescription or a best practice?
This is a best practice recommendation to limit spraying insecticides and conserve natural enemies at the early stages of the cropping season
- Section 2.3– were cell phones not utilized for the promotional rollout? My understanding is that diffusion of cell phones is fairly robust, even in rural areas, and often used by Extensionists (at least in the Philippines, would assume Vietnam is similar).
Cellphones were used but this was not a specific component of the campaign and roll-out managed by the government.
- LINE 257Extra space before Moreover
Removed the extra space.
- LINE 459“Notably, 3R3G has some of the components of 1M5R” Can commonalities be outlined in a table? OR include a summary timeline/chronology of the progenitor and successor systems (starting with TRTG and ending with “beyond” 1M5R) with associated years and core practices - would be a useful addition to observe the evolutionary continuum.
This is addressed in the previous comment. The overlaps are outlined in the revised Table 1.